# Protocol of a parallel group Randomized Control Trial (RCT) for Mobile-assisted Medication Adherence Support (Ma-MAS) intervention among Tuberculosis patients

Zekariyas Sahile[1,2]*, Lua Perimal-Lewis[3], Paul Arbon[4], Anthony John Maeder[2]

**1** Department of Public Health, Ambo University, Ambo, Ethiopia, **2** Flinders Digital Health Research Centre College of Nursing & Health Sciences, Flinders University, Adelaide, SA, Australia, **3** College of Science & Engineering, Flinders Digital Health Research Centre, Flinders University, Adelaide, SA, Australia, **4** College of Nursing & Health Sciences, Flinders University, Adelaide SA, Australia

\* zekariyas.nezenega@flinders.edu.au, zekitiru@gmail.com

## Abstract

### Background

Non-adherence to Tuberculosis (TB) medication is a serious threat to TB prevention and control programs, especially in resource-limited settings. The growth of the popularity of mobile phones provides opportunities to address non-adherence, by facilitating direct communication more frequently between healthcare providers and patients through SMS texts and voice phone calls. However, the existing evidence is inconsistent about the effect of SMS interventions on TB treatment adherence. Such interventions are also seldom developed based on appropriate theoretical foundations. Therefore, there is a reason to approach this problem more rigorously, by developing the intervention systematically with evidence-based theory and conducting the trial with strong measurement methods.

### Methods

This study is a single-blind parallel-group design individual randomized control trial. A total of 186 participants (93 per group) will be individually randomized into one of the two groups with a 1:1 allocation ratio by a computer-generated algorithm. Group one (intervention) participants will receive daily SMS texts and weekly phone calls concerning their daily medication intake and medication refill clinic visit reminder and group two (control) participants will receive the same routine standard treatment care as the intervention group, but no SMS text and phone calls. All participants will be followed for two months of home-based self-administered medication during the continuation phases of the standard treatment period. Urine test for the presence of isoniazid (INH) drug metabolites in urine will be undertaken at the random point at the fourth and eighth weeks of intervention to measure medication adherence. Medication adherence will also be assessed by self-report measurements using the AIDS Clinical Trial Group adherence (ACTG) and Visual Analogue Scales (VAS) questionnaires, and clinic appointment attendance registration. Multivariable regression model analysis will be employed to assess the effect of the Ma-MAS intervention at a significance level of P-value < 0.05 with a 95% confidence interval.

**Funding:** This trial protocol is partially funded by Flinders University. The funder and sponsor had no role in trial design, and will not have a role in the

trial implementation, analysis, and interpretation of the findings. All trial information is the responsibility of the authors.

**Competing interests:** The authors declared that they have not any competing interests.

## Discussion

For this trial, a mobile-assisted medication adherence intervention will first be developed systematically based on the Medical Research Council framework using appropriate behavioural theory and evidence. The trial will then evaluate the effect of SMS texts and phone calls on TB medication adherence. Evidence generated from this trial will be highly valuable for policymakers, program managers, and healthcare providers working in Ethiopia and beyond.

## Trial registration

The trial is registered in the Pan-Africa Clinical Trials Registry with trial number PACTR202002831201865.

## Background

Non-adherence to Tuberculosis (TB) treatment is a risk factor for further transmission, treatment failure, relapse, acquired multi-drug resistance, or extensively drug-resistant TB and death [1–3]. In Ethiopia, non-adherence to TB treatment, specifically a medication regimen involving a combination of antibiotics, is a serious threat to TB prevention and control programs. A recent systematic review and meta-analysis conducted in Ethiopia found that the pooled prevalence of non-adherence to TB medication was 21.3% [4].

Non-adherence to TB treatment is influenced by many interconnected factors. These include patient-centred-related factors, economic-related factors, social-related factors, health system-related factors, therapy-related factors, geographical access-related factors and lifestyle-related factors [5, 6]. The major factors affecting TB treatment adherence in Ethiopia are patient-centred-related factors such as knowledge about TB, its treatment duration and consequence of non-adherence [7–9], educational status [7, 10], psychological state of the patients like forgetfulness [10–13] and psychological distress [7, 14] and social-related factors such as social support, stigma and discrimination, perceived risk, perceived wellness/cure, and perceived barriers over the benefits [7, 14].

The World Health Organization (WHO) recommended Direct Observation of Treatment (DOT) by a trained supervisor; a health worker or a treatment supporter watches the patient take their antibiotics every day to ensure adherence [15]. However, evidence shows that DOT does not necessarily provide a reliable solution to poor TB treatment adherence [16, 17]. In Ethiopia, implementing DOT is also very challenging for patients as it requires patients' daily visits to the health facility, which has undesired implications in their work, social life and high transportation cost [18, 19]. The current WHO recommendations are to consider DOT implementation with different administration options including conventional DOT, video-observed treatment (VOT), and non-daily DOT [20]. The non-daily DOT has been implemented in Ethiopia, in which during the first two months of the intensive phase of treatment patients take their daily medications with the supervision of the healthcare provider, and from two to six months in the continuation phase of treatment patients take all their medications at home and visit a clinic weekly to refill their medications [11, 19]. However, the non-daily DOT schedule has been interrupted by the COVID-19 pandemic [21].

Digital health interventions through mobile health can create new opportunities for healthcare improvements, especially in situations with limited infrastructure, expertise, and human

resources in the healthcare system [22–25]. Digital health interventions that enhance TB medication adherence are seen as a promising option to bring health benefits to the community [26]. The growth in popularity of mobile phones provides opportunities to address adherence challenges related to DOT, as mobile health can facilitate direct communication between healthcare providers and TB patients through SMS and voice calls along with DOT or non-daily DOT [27, 28]. Along with SMS text reminders, phone calls are also suggested to improve a patient's clinic attendance and treatment outcomes, and can also be used to make the SMS text intervention interactive [29, 30]. In Ethiopia, access to mobile phones is expanding widely: according to the Ethiopia Demography Health Survey (EDHS) 2016, 88% of urban households and 47% of rural households have mobile phone access [31]. This is substantial growth over 5 years when compared to the EDHS 2011, which indicated 65% of urban households and 13% of rural households had mobile phone access [32].

There is some evidence that interventions that rely on communications, like educating and counselling the patient, or medication monitoring and reminders, can improve treatment adherence and success [28, 33–35]. In Ethiopia, for example, a randomized trial in 2016 found that psychological counselling and educational intervention can significantly improve a patient's treatment adherence [36]. Another trial in Ethiopia in 2019 has investigated the effect of a daily SMS text and weekly medication refill reminders on patient's medication adherence [37]. However, the combination of SMS text and phone calls (as in the intervention proposed here) based on the application of formal behavioural theories and behavioural techniques has not been previously designed and tested. This trial will also apply a combination of direct and indirect methods of adherence measurements for outcomes assessment. The mobile-assisted medication adherence support intervention will be developed systematically based on the Medical Research Council model, incorporating appropriate theory and evidence. The intervention design will be informed through qualitative in-depth interviews with different stakeholder groups such as patients, healthcare providers and healthcare managers, and the SMS content will be validated with experts in behavioural science, healthcare system, pulmonary disease and digital health, using a Delphi technique and will be revalidate by health professionals working at ground level.

## Methods

### Hypothesis

We plan to test whether our Mobile-assisted Medication Adherence Support (Ma-MAS) intervention using mobile SMS text reminders and phone calls will improve Tuberculosis medication adherence in addition to routine standard non-daily DOT care.

### Trial design

A parallel-group individual randomized controlled trial (RCT) with two groups will be employed to investigate the effect of Ma-MAS intervention on TB medication adherence. Participants will be individually randomized into one of the two groups (one intervention and one control group) in a 1:1 allocation ratio (Fig 1).

### Study setting

The study will be conducted at the primary public health facilities of Addis Ababa, Ethiopia. Addis Ababa is the capital city of Ethiopia, and the city has 11 administrative sub-city regions. Based on population projection the estimated total population of Addis Ababa in 2017 was 3,433,999 with 1,624,999 Males and 1,809,000 Females [38]. In Ethiopia, a DOT program has

| Study Period | | | | | |
|---|---|---|---|---|---|
| | Enrolment | Allocation | Post-allocation | | close-out |
| **Timeline** | - t1 | t0 | t1 | t2 | t3 |
| | Week 1 | Week 12 | Week 16 | Week 20 | week 21 |
| **Enrolment** | | | | | |
| Eligibility screening | ✖ | ✖ | | | |
| Informed consent | ✖ | ✖ | | | |
| Allocation | ✖ | ✖ | | | |
| **Intervention** | | | | | |
| Daily SMS text | | | ←———————→ | | |
| Weekly Phone calls | | | ←———————→ | | |
| **Assessment** | | | | | |
| **Baseline** | | | | | |
| Socio-demographic | ✖ | ✖ | | | |
| Clinical characterstics | ✖ | ✖ | | | |
| Substance Use-related information | ✖ | ✖ | | | |
| Health Service-related information | ✖ | ✖ | | | |
| Adherence measured by AIDS Clinical Trial Group adherence questionnaire | ✖ | ✖ | | | |
| Adherence measured by Visual Analogue Scales (VAS) | ✖ | ✖ | | | |
| **Outcome** | | | | | |
| Adherence measured by IsoScreen test | | | | ✖ | ✖ |
| Adherence measured by AIDS Clinical Trial Group adherence questionnaire | | | | | ✖ |
| Adherence measured by Visual Analogue Scales (VAS) | | | | | ✖ |
| Adherence measured by clinic appointment attendance | | | ←———————→ | | |

**Fig 1. Timeline for Ma-MAS intervention trial (SPIRIT figure).**

been implemented in two phases, "intensive" and "continuation". In the intensive phase (daily DOT), a patient takes their medication in front of the healthcare worker every day at a health facility. In the continuation phase (non-daily DOT), a patient is given the responsibility to take

the medication at home and come to a health facility weekly for health checking and medication refilling [39].

## Participants

Adult Tuberculosis patients who are receiving anti-TB treatment at primary public health facilities of Addis Ababa, Ethiopia will be prospective participants for the trial.

## Eligibility

**Inclusion criteria.**   To be included in the trial:

- participants must be enrolled in a primary public health facility for anti-TB treatment and have attained their first two months of the intensive phase of treatment

- participants must be aged 18 years and above

- participants must be able to read and understand SMS text that is written in the national official language (Amharic) of Ethiopia

- participants must have their own mobile phone, or

- participants who do not have a mobile phone can be included if they have a shared mobile phone in the household with a collaborative agreement; (i.e. a voluntary agreement between the patient and family member living in the same household).

**Exclusion criteria.**   Participants will be excluded from the trial under the following conditions:

- participants whose anti-tuberculosis treatment has been underway for more than six months.

- participants who are enrolled or have agreed to enroll in any other interventional study at the same time as this study is conducted.

## Intervention group

The intervention will be given in arm-1 (intervention) group participants will receive a daily SMS text and weekly phone calls for medication intake and medication refill visit reminders. All participants will receive the intervention at a similar time for two months during the continuation phase of anti-TB treatment (Fig 2).

## Control group

The control group (Arm-2) participants will receive the same routine standard treatment care as the intervention group, but no SMS text and phone calls intervention. Patients in the control group will be followed for the same two-month period as the intervention group during the continuation phase of their treatment.

## Intervention development procedure

The mobile SMS text messages will be developed systematically with the best available evidence in the context area and with the application of appropriate behavioural theory and technique. The intervention will be developed and evaluated based on the Medical Research Council

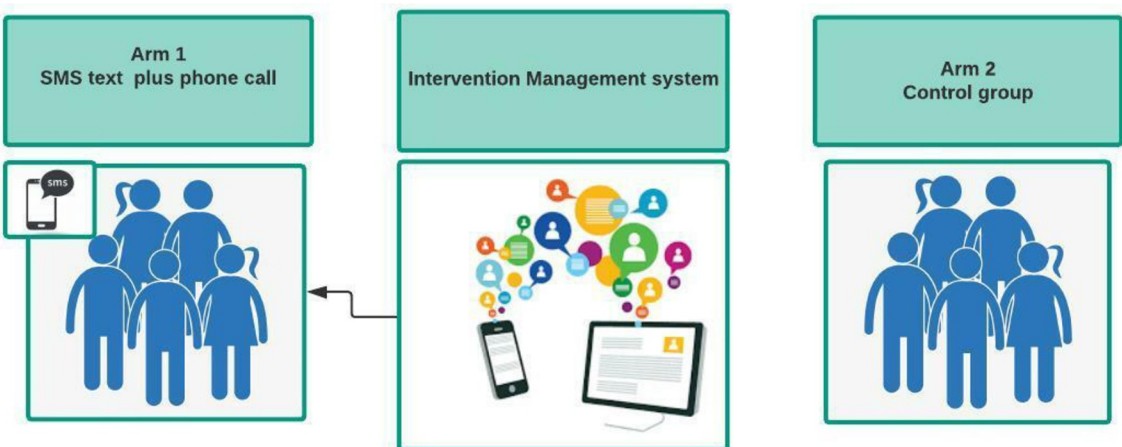

**Fig 2. Mobile-assisted Medication Adherence Support (Ma-MAS) intervention diagram.**

(MRC) framework for complex interventions in healthcare (Table 1) [40]. The non-sequential development stage of the framework as adapted by Bleijenberg et al. (2018) will be used as a more specific guide for the intervention design [41]. This has seven different phases:

Phase 1: problem definition (identification and analysis)

Phase 2: systematically identifying the evidence

Phase 3: identifying and developing a theory

Phase 4: determining the needs (patients and/or providers)

Phase 5: examining the practice

Phase 6: modeling process and outcome and

Phase 7: intervention design.

**Table 1. Mobile-assisted medication adherence support intervention development process using an adapted development phase of MRC framework and the associated methods.**

| Development phases | Methods |
|---|---|
| Phase 1: Problem identification and definition | • Review of literature<br>• Qualitative research |
| Phase 2: Identifying the evidence base | • Review of literature<br>• Qualitative research<br>• Quantitative Survey research |
| Phase 3: Identifying and/or developing theory | • Review of literature |
| Phase 4: Determine the need | • Review of literature<br>• Qualitative research |
| Phase 5: Examine current practice and context | • Qualitative research |
| Phase 6: Modeling process and outcome | • Modeling a prototype of intervention components<br>• Identity interrelation with the outcome<br>• Experts opinion through Delphi method and survey |
| Phase 7: Intervention design | • Experts opinion<br>• Refine the intervention |

**Phase 1: Problem identification and definition.**   The problem of non-adherence to the anti-TB medication has been identified and defined through a literature review. The literature review is published in the title of "Factors Influencing Patient Adherence to Tuberculosis Treatment in Ethiopia? A Literature Review" [6]. This review will be further supported by qualitative research using in-depth interviews with TB patients and healthcare providers.

**Phase 2: Identifying the evidence base.**   Evidence has been collected through a review of literature, to identify the effectiveness of mobile health SMS interventions including acceptability, feasibility, and characteristics of the intervention. Acceptability and feasibility survey will be conducted before the trial is implemented. This phase aims to identify previous interventions and their effectiveness and to assess the acceptability and feasibility of the target groups of the interventions, as well as the success and failure factors (S1 and S2 Tables).

**Phase 3: Identifying and/or developing theory.**   The theories or models that can best explain the proposed intervention mechanism and action have been selected through a review of the literature. This phase of development aims to identify the theory or model that can best explain the intervention mechanism of action. We have selected an information motivation behavioural skill model [42], and have shown the relationship between the model construct of information, motivation and behavioural skills with adherence behaviour based on the factors affecting TB medication adherence identified in phase one (Fig 3).

**Phase 4: Determine the needs.**   The needs, preferences, perceptions, and capacities of the patient and healthcare providers with the identified problems and proposed solutions will be determined through a review of the literature and qualitative research by using in-depth interviews. This will generate information about which characteristics of the intervention are likeable and adaptable, and how the SMS texts can be tailored regarding language, content, frequency, length, and time of the day.

**Phase 5: Examine current practice and context.**   Identification of barriers and facilitators of mobile health interventions through examining the current practice and context from the patient's and healthcare provider's perspectives will be made. This would help the functionality of the proposed mobile health intervention best fits with the existing practice and context. This activity will be done through qualitative research by using in-depth interviews.

**Phase 6: Modelling process and outcome.**   Modeling the active component of the mobile health intervention and its relationship with outcomes will be refined with multidisciplinary team involvement. The initial SMS text content will be refined using the taxonomy of evidence-based behaviour change [43]. The Delphi technique will be employed to validate the content of the SMS text further. Eight experts from different professional backgrounds of behavioural science, healthcare system, pulmonary disease, and digital health experts will be involved in this process. A survey will be conducted among health professionals who working as TB focal persons and communicable disease control officers from health facilities and health offices in Addis Ababa, Ethiopia.

**Phase 7: Intervention design.**   A full prototype mobile SMS text intervention will be developed and will be given to experts for their feedback to make final refinements. The experts will be asked to comment on the timing and frequency of information, clarity, and arrangement of the SMS text messages. Based on the expert's comments the final SMS text message set will be prepared.

## Outcome

The outcome of the trial is the medication adherence of the patient, measured by both direct and indirect methods of measurement. The primary outcome will be measured using an Iso-Screen test (GFC Diagnostics Ltd, Bicester, England) for the presence of isoniazid (INH) drug

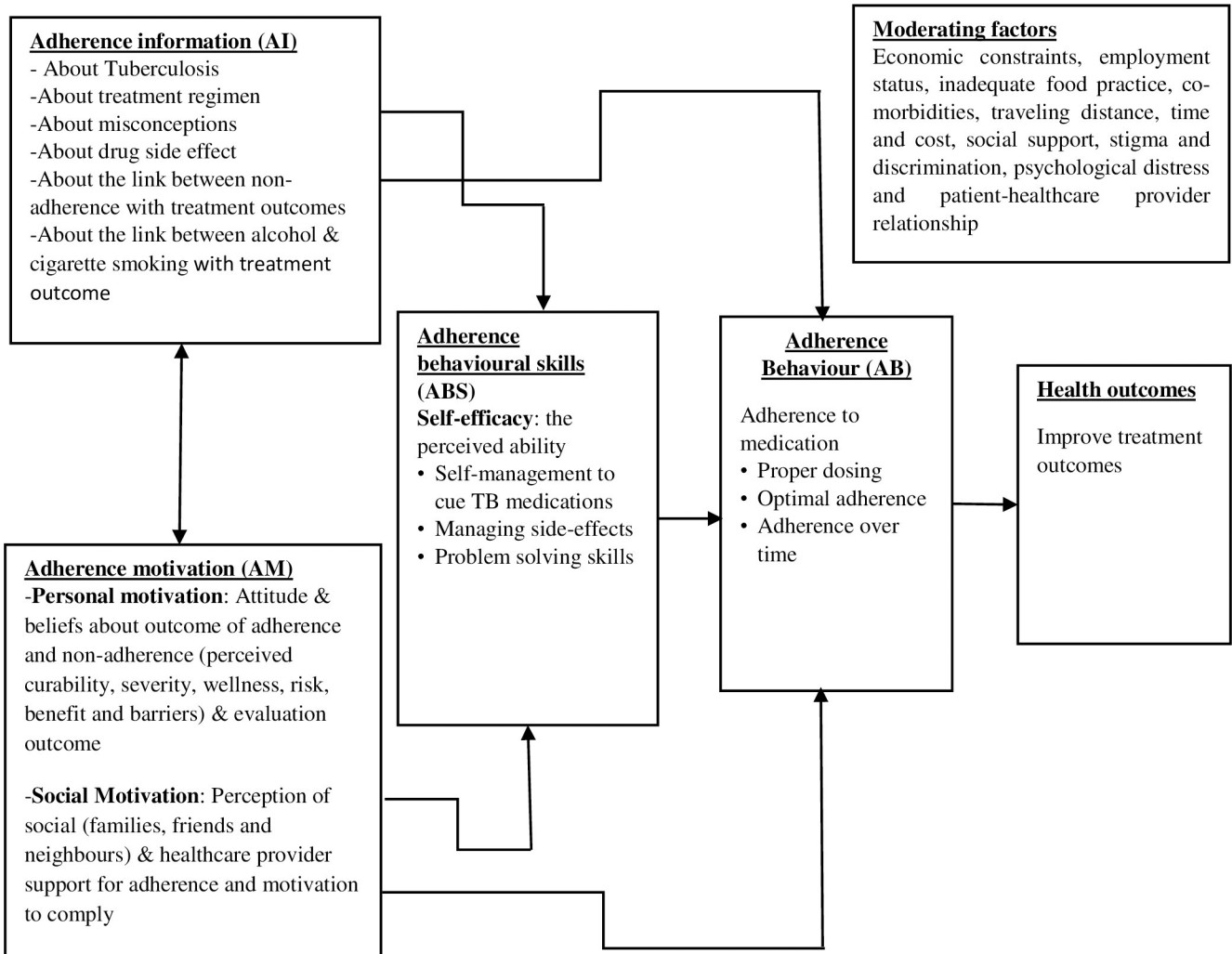

**Fig 3. Information, motivation and behavioural skills model to Tuberculosis medication adherence adapted from Fisher et al. 2006 [42].**

metabolites in urine in accordance with the vendor-supplied user manual. The IsoScreen test result is negative when there is no colour change (i.e. the strip remains yellow) after 5 minutes, and the test result is positive when the colour changes to dark purple, or blue/ green in that time, depending on the concentration of the medication (i.e. if the medication was taken in the last 24 hours or 48 hours respectively) [44, 45]. The IsoScreen test will be done at the fourth and eighth weeks of the intervention. The proportion of adherence measured by the IsoScreen test at the eighth week of the intervention (primary end point) will be used as the primary interest of analysis. The proportion of adherence measured by the IsoScreen test at the fourth week of the intervention will be used as the secondary interest of analysis.

The secondary outcome will also be measured by self-report of medication adherence using the adapted AIDS Clinical Trial Group adherence questionnaire (ACTG) [46], Visual Analogue Scales (VAS) [47], and by clinic appointment attendance registration [48]. The adapted version of ACTG has three sections: section B (social support), section C (possible reasons for non-adherence) and section D (adherence behaviour). Participant scores less than "somewhat satisfied" to any questions in section B or other than "never" to any questions in sections C

and D are considered as non-adherence [48]. VAS is a single adherence scale that requires a patient to estimate their adherence level from 0% to 100%: 0% means the patient has taken no medication, 50% means the patient has taken half of their prescribed medications, and 100% means the patient has taken all of their prescribed medications, for the last 30 days. Participant scores less than 90% for VAS are considered as non-adherence [36, 49]. Clinic appointment attendance registration for weekly medication refills visits will also be used for adherence measurement. Patients who delay for at least one medication refill visit are considered as non-adherence [48]. The ACTG and VAS questions will be asked at bassline assessment and end of two months of intervention (Fig 1). The proportion of adherence measured by self-report and clinic appointment attendance registration at the eighth week of the intervention will be used as the secondary interest of analysis.

## Sample size

A STATA code *power twoproportions* was used to calculate the sample size considering the trial outcome of anti-TB medication non-adherence. The rate of anti-TB medication non-adherence in Ethiopia varies considerably in the literatures, ranging from 10% to 26% depending on the patient's treatment phase and different settings [8, 11, 14]. It was found specifically that the rate of non-adherence was 25.6% (po = 0.256) during the continuation phase of treatment from a recent study conducted at Addis Ababa, Ethiopia [14] in the control group. We used a 15% absolute reduction of non-adherence due to a mobile SMS text intervention found in a similar study [37] to estimate a non-adherence rate of 10.6% (pa = 0.106) in the intervention group. Considering that we are using a simple randomized controlled trial parallel-group design, choosing a 95% confidence level (alpha = 0.05), 80% power (beta = 0.80) and a one-sided p-value of 0.025 for the initial sample size would yield n = 81 participants per arm. Assuming an attrition rate of 15%, the final sample size would be a total of n = 186 with each of the 2 arms having n = 93. From 10 sub-cities in Addis Ababa, a total of 36 primary public health facilities will be included in the trial study by random selection, based on the assumption that on average 6 patients per public health facility will be available to be included in the study.

## Recruitment

Patients enrolled in the intensive phase of anti-TB treatment will be recruited by the research support team of healthcare professionals from each health facility, based on the eligibility and exclusion criteria. The recruitment will be continued until the required sample size is obtained. All the recruitment process details will be recorded on paper forms and kept securely by the facilities until the project is completed.

## Randomization and allocation

Participants will be centrally randomized into one of the two groups using simple randomization in computer-generated algorithm, with an equal allocation ratio (1:1), by members of the research support team, who will not be involved in measuring the outcome (Fig 4).

## Follow up

Both the intervention and control groups' patients will be followed for an equal period of 2 months. Intervention group participants will receive a total of 60 daily SMS text reminders and 8 phone calls for medication intake and refill visits reminders.

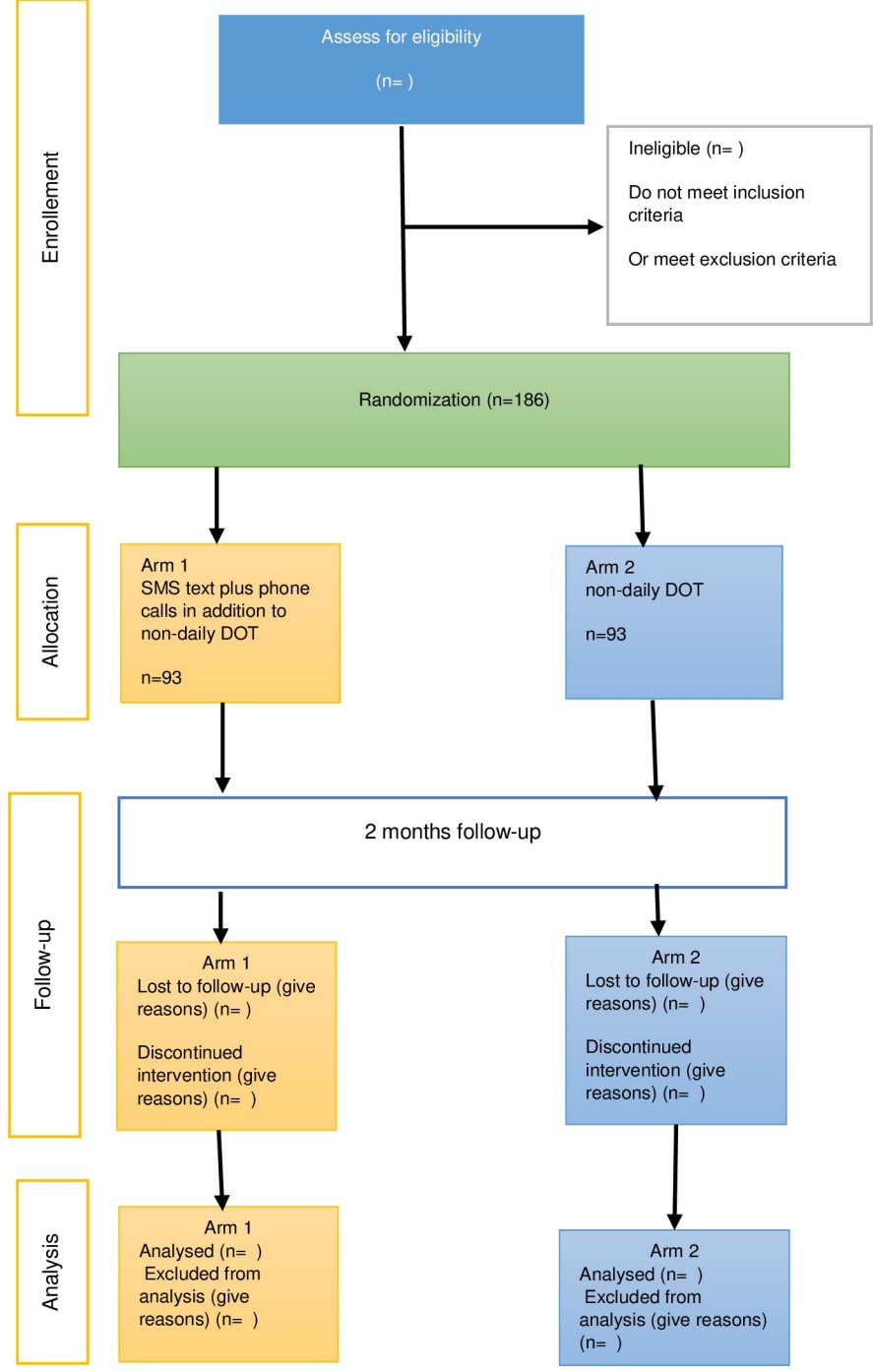

**Fig 4. Flow diagram for Ma-MAS intervention trial.**

## Blinding

Single-blinding will be applied to avoid bias in outcome measurement, by differentiating the person who measures the outcome from the person who randomizes and provides the intervention. Thus, the person who measures the outcome will be blinded as to which group

participants are assigned. Due to the nature of the trial intervention participants will not be blinded to which group they are assigned.

## Implementation

All patients assigned to any group will get the same standard treatment based on the Ethiopia National Tuberculosis Treatment guideline which is rifampicin, isoniazid, pyrazinamide and ethambutol for two months followed by rifampicin and isoniazid for four months [50]. Patients screening, enrolment, consenting and randomization will be undertaken by the research support team, who have health professional backgrounds. An open SMS automation software platform will be used to store, monitor, and send SMS texts. Global system for mobile (GSM) sim card and short code SMS credit will be used to send the SMS texts. SMS text traffic will be managed and monitored in one centralised computer database server by trained IT personnel. The maximum size of an SMS text will not be more than 160 characteristics. A weekly phone calls will be made to the intervention group participants from the health facility by delegated members of the research team. A checklist of questions will be recorded at each time the weekly phone calls are made to the participants (S2 Checklist). Bothe the SMS text and Phone calls intervention will be made using the national official language which is Amharic. For patients who do not answer phone calls, three repetitive attempts will be made at five-minute intervals. All SMS texts and phone calls will be delivered at a similar time in the morning before the actual medication time. The message delivery reports will be continuously monitored to ensure the participants have received the SMS texts as planned. Participants will also be asked whether they are receiving the SMS text every week through phone calls and when they come for their medication refill visit. Participants will be briefed on the duration, frequency, and timing of SMS text and phone calls intervention and advised to turn on their mobile in the morning. Participants will be informed about the intervention and instructed not to tell and/or share the SMS texts or phone calls with other people. If participants lose or change their phone numbers, the new phone number will be replaced immediately after they have informed a support team member. The trial will be audited with participant enrolment, consent, allocation, and during the intervention period.

## Data collection and management

A trained healthcare professional team will collect and supervise the data collection. Data collection will be made after the patient has received the routine non-daily DOT services in a different room from the TB clinic room, to ensure the confidentiality and privacy of the patient's information. The data will be collected electronically using a smart mobile app through a questionnaire coded using Open Data Kit (ODK). Data will be collected from the primary source through interviews and sample urine collection, and secondary source by a review of the health facility attendance registration book to confirm the patient's medication refill visit attendance. All interviews will be conducted after the patients finish the non-daily DOT service. Interviews will be conducted in a different room from the TB clinic to be convenient for discussion and privacy.

To ensure the security of the patients' information, all data will keep in Open Data Kit (ODK) database, principal investigator laptop computer, and backup file with password protectors. All identifiable information will be removed and replaced by code. All hard-copy research documents will be locked in a cabinet.

## Statistical analysis

The electronically collected data will be exported to STATA and cleaned using STATA coding programing language. Descriptive statistics of mean, standard deviation, median, and range will be used for continuous variables and frequency and percentage for categorical variables.

Intention to treat (ITT) analysis will be performed for all randomized participants according to their group. A chi-square test for binary outcomes and Student's t-test for continuous outcomes will be used as appropriate. For primary and secondary interest of analysis, multivariable regression model analysis will be employed to adjust confounding factors in the statistical effect. A P-value of less than 0.05 will be used as a cut-off point to show a statistically significant association between Ma-MAS intervention and medication adherence with a 95% confidence interval.

## Ethical considerations

The research proposal has been approved by the Southern Adelaide Clinical Health Research Ethics committee (SAC HREC) on April 8, 2020, with Office for Research (OFR) number: 23.20. The proposal has also be approved by the Addis Ababa Health Bureau ethics committee on August 21, 2020, with reference number A/A/H/930/227 and an official permission letter is provided from Addis Ababa Health Bureau to respective sub-city and health facilities. Each participant will complete a written informed consent form after being informed about the research objectives, procedures, anticipated risks, and benefits of the research. The research team will be confirming or renegotiating consent with participants for additional data. The principal investigator will be ensuring participants are confirming their consent to participation. Participants have the right to withdraw from the study at any time when they feel uncomfortable or unable to continue in the study, and they will be informed of its limitations or consequences on the research during consent requests. Any change in the study protocol and participant informed consent will be submitted to SAC HREC before implementing the change. This trial has no forms of inducement, coercion and the study does not bring any risks that incur compensation in cash or kind.

## Publication and dissemination of the result

The trial findings will be reported based on the Consolidated Standard of Reporting Trials (CONSORT) guidelines. The research outcomes will be published in an international peer-reviewed open access journal. The research outcomes will also be presented at selected national and international conferences. The principal investigator is responsible for disseminating the finding to Health facilities, Addis Ababa Health Bureau, Federal Ministry of Health (FMOH), Flinders University, national and international conferences. The participants will be asked whether they wish to get a summary of the research work, and if so in what form, and will be informed about how they can access the research results. In the case of a formal request for further research purposes, non-identifiable secondary data may be granted to others. A participant's identifiable information will be removed and replaced by code to protect the anonymity of the participants. Identifiable data and/or information will not be accessed or granted to others.

The trial is registered in the Pan-Africa Clinical Trials Registry with trial number PACTR202002831201865.

## Discussion

An adherence intervention needs to address the underlying factors that cause medication non-adherence [26]. Medical Research Council (MRC) guidelines recommend using appropriate theory and evidence when developing an intervention [40]. However, most adherence interventions are developed without a strong theoretical base, which might be one of the reasons they have not been effective [51]. In this randomized controlled trial, the Mobile-assisted

Medication Adherence Support (Ma-MAS) intervention will be developed systematically using the MRC conceptual framework with an appropriate selection of behavioural theory.

Some evidence has shown that interactive SMS texts may have a better impact than using simple SMS text reminders in improving medication adherence [52, 53] although there is also inconsistent evidence found in some two-way [30, 54] and one-way SMS text studies among TB patients [55, 56]. This trial was designed to test an intervention with a combination of daily SMS text and weekly phone calls for medication intake and medication refill visit reminders during the continuation phase of treatment for a two months period, which was not studied previously.

When testing an intervention with randomized trials, correctly measuring outcomes in the same way for both the intervention and control groups is crucial. The accuracy of adherence measurement is very important to assess the true effect of an adherence intervention. Here the adherence outcome will be measured through the combination of a direct method (IsoScreen test) and several indirect methods (VAS, ACTG, and clinic attendance registration) of measurement. This will help to avoid bias in outcome measurement. Therefore, this research would provide evidence to help healthcare managers and policymakers to develop strategies that improve Tuberculosis medication adherence and treatment outcomes in general.

## Supporting information

**S1 Checklist. SPIRIT 2013 checklist: Recommended items to address in a clinical trial protocol and related documents**[*].
(DOC)

**S2 Checklist. English version checklist for phone calls intervention.**
(PDF)

**S1 Table. SMS text messaging interventions for TB treatment support.**
(DOCX)

**S2 Table. SMS interventions for TB: Acceptability and feasibility study design.**
(DOCX)

**S1 File. Participant information sheet/consent form intervention and post-intervention participants.**
(PDF)

**S2 File. Mobile-assisted Medication Adherence Support (Ma-MAS) intervention for Tuberculosis patients: Adoption of intention, effectiveness, and experiences in Addis Ababa, Ethiopia.**
(PDF)

**S3 File. Pan African Clinical Trials Registry.**
(PDF)

## Acknowledgments

We would like to thank the Addis Ababa Health Bureau and Federal Ministry of Health of Ethiopia for facilitating the necessary conditions to conduct the trial. We would also like to thank Shahid Ullah from Flinders University, College of Medicine and Public Health, for advising us on sample size calculation.

## Author Contributions

**Conceptualization:** Zekariyas Sahile, Lua Perimal-Lewis, Paul Arbon, Anthony John Maeder.

**Funding acquisition:** Zekariyas Sahile, Anthony John Maeder.

**Methodology:** Zekariyas Sahile, Lua Perimal-Lewis, Paul Arbon, Anthony John Maeder.

**Project administration:** Zekariyas Sahile.

**Supervision:** Lua Perimal-Lewis, Paul Arbon, Anthony John Maeder.

**Writing – original draft:** Zekariyas Sahile, Lua Perimal-Lewis, Paul Arbon, Anthony John Maeder.

**Writing – review & editing:** Zekariyas Sahile, Lua Perimal-Lewis, Paul Arbon, Anthony John Maeder.

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
