## [Decision Letter · Decision Letter 0]

9 Aug 2021

PONE-D-21-07609

Protocol of a Parallel Group Randomized Control Trial (RCT) for Mobile-assisted Medication Adherence Support (Ma-MAS) intervention among Tuberculosis Patients

PLOS ONE

Dear Dr. Sahile,

Thank you for submitting your manuscript to PLOS ONE. After careful consideration, we feel that it has merit but does not fully meet PLOS ONE’s publication criteria as it currently stands. Therefore, we invite you to submit a revised version of the manuscript that addresses the points raised during the review process.

The study is of significance and fits within the journal scope, however the manuscript needs further details to be suitable for publication.  Please see the comments below from the reviewers, in particular the drug types/scheduling details for the intervention and control groups need to be more specific.

In addition, consider whether Figure 2. needs modifying, or whether is necessary, as at present it indicates an arrow from the intervention section to the control group, which is somewhat confusing as I am assuming the control group do not receive the intervention. 

We look forward to receiving your revised manuscript.

Kind regards,

Kathleen Finlayson

Academic Editor

PLOS ONE

Journal Requirements:

Reviewers' comments:

Reviewer's Responses to Questions

**Comments to the Author**

1. Does the manuscript provide a valid rationale for the proposed study, with clearly identified and justified research questions?

Reviewer #1: Yes

Reviewer #2: Yes

2. Is the protocol technically sound and planned in a manner that will lead to a meaningful outcome and allow testing the stated hypotheses?

Reviewer #1: Partly

Reviewer #2: Yes

3. Is the methodology feasible and described in sufficient detail to allow the work to be replicable?

Reviewer #1: Yes

Reviewer #2: Yes

4. Have the authors described where all data underlying the findings will be made available when the study is complete?

Reviewer #1: Yes

Reviewer #2: No

5. Is the manuscript presented in an intelligible fashion and written in standard English?

Reviewer #1: Yes

Reviewer #2: Yes

6. Review Comments to the Author

You may also provide optional suggestions and comments to authors that they might find helpful in planning their study.

Reviewer #1: Comments and Suggestions for Authors

1. What type of medicine is the patient taking? Is it fixed dosed combination (FDC) or combination drugs with individual strips for each drug? because if the patient is taking the last type, maybe urine tests with the measurement of the metabolite Isoniazid don't seem to be able to represent patients’ adherence because Isoniazid and Rifampicin are taken once a day while Pyrazinamide and Ethambutol are taken 3 times a day for intensive treatment phase.

2. It will be better if the authors provide the reason or evidence why the adherence test with the urine test will be tested on the fourth and eighth weeks of intervention. Also, provide evidence that urine test has superiority to measure adherence than other methods.

3. The method how to measure adherence in the abstract and the body text seems not consistent. Please make it consistent. The abstract has only been written with urine test but in the body text, there are some measurements: urine test, ACTG, VAS, and clinical attendance registration.

4. How to interpret the result of the urine test? The author stated that the test result is positive when the colour changes to dark purple, or blue/ green at that time, depending on the concentration of the medication (i.e. if the medication was taken in the last 24 hours or 48 hours respectively). So if the patient was taking drugs two days ago (in the last 48 hours before the urine test), the result will be blue/green and will be categorized as positive and adherence? Shouldn't in the intensive phase of treatment the drugs have to be taken every day?

5. Intervention by SMS or telephone depending on the signal and the mobile phone is on. The author mentions that if there is no response, the call will be repeated 3 times and an SMS will be sent in the morning. It is necessary to inform and advise the patient to turn on the mobile phone in the morning so that the message or call can be received by the patient.

6. The authors used several methods to assess patient adherence. What if there are discrepancies in the results of the adherence examination using a urine test, Clinical Trial Group adherence questionnaire (ACTG), Visual Analogue Scales (VAS), and clinical attendance registration? For example, the result of one of the tests is categorized as non-adherence and the other tests are categorized as adherence or vice versa?

7. Maybe authors can add Alternative analysis for student T-tests if the data is not normally distributed.

Reviewer #2: It is unclear which is the primary outcome of the trial, and how it will be measured. Please be more specific. Please classify only one measure as the primary outcome (e.g. compliance as measured by IsoScreen) and others as secondary (e.g. compliance as measured by self report and clinic attendance.)

For the IsoScreen test, when will the test be done?

When will the ACTG adherence questionnaire be administered? Will this be considered as a binary outcome (adherent/nonadherent) or continuous (50%, 65%, etc?)

7. PLOS authors have the option to publish the peer review history of their article (what does this mean?). If published, this will include your full peer review and any attached files.

Reviewer #1: No

Reviewer #2: No

---

## [Author Response · Author response to Decision Letter 0]

19 Aug 2021

Dear Sirs 

We would like you thank you for comments. We have seen the reviewers' comments carefully one by one. The comments are essential, and we revised the protocol manuscript as per the reviewers' and editor's comments. We attached the response to reviewers comments document. We hope all comments are addressed in the revised protocol manuscript. Thank you.

---

## [Decision Letter · Decision Letter 1]

1 Oct 2021

PONE-D-21-07609R1Protocol of a Parallel Group Randomized Control Trial (RCT) for Mobile-assisted Medication Adherence Support (Ma-MAS) intervention among Tuberculosis PatientsPLOS ONE

Dear Dr. Sahile,

Thank you for submitting your manuscript to PLOS ONE. After careful consideration, we feel that it has merit but does not fully meet PLOS ONE’s publication criteria as it currently stands. Therefore, we invite you to submit a revised version of the manuscript that addresses the points raised during the review process. In particular,  please clarify the primary outcome and randomization definitions and timing as per the reviewer comments, and address whether a clustering effect should be allowed for in the sample size calculations.

We look forward to receiving your revised manuscript.

Kind regards,

Kathleen Finlayson

Academic Editor

PLOS ONE

Journal Requirements:

Additional Editor Comments (if provided):

Reviewers' comments:

Reviewer's Responses to Questions

**Comments to the Author**

1. Does the manuscript provide a valid rationale for the proposed study, with clearly identified and justified research questions?

Reviewer #1: Yes

Reviewer #2: Yes

2. Is the protocol technically sound and planned in a manner that will lead to a meaningful outcome and allow testing the stated hypotheses?

Reviewer #1: Yes

Reviewer #2: Yes

3. Is the methodology feasible and described in sufficient detail to allow the work to be replicable?

Reviewer #1: Yes

Reviewer #2: Yes

4. Have the authors described where all data underlying the findings will be made available when the study is complete?

Reviewer #1: Yes

Reviewer #2: No

5. Is the manuscript presented in an intelligible fashion and written in standard English?

Reviewer #1: Yes

Reviewer #2: Yes

6. Review Comments to the Author

You may also provide optional suggestions and comments to authors that they might find helpful in planning their study.

Reviewer #1: The outcome of this study is only one outcome, namely adherence. This outcome will be measured by using several methods or measurement tools. Can one outcome (adherence) be a primary and secondary outcome? The authors may consider using “outcome (adherence) will be measured by direct measurement (urine test) and indirect measurement (ACTG and VAS)”, without “primary and secondary outcome”. In my opinion, using “primary and secondary outcome” but the outcome is only one (adherence) is confusing.

Reviewer #2: 1. It is unclear if the primary outcome is the ISOCreen done at 4 or 8 weeks of the intervention.

2. Is clustering by health facility an issue? If yes, then the sample size should account for this as should the statistical analysis. If not, then a rationale should be provided for why the outcome does not vary by health facility.

3. Randomization: The randomization described seems to be stratified by health facility. Please update the description accordingly. Also , is a simple randomization planned? What about blocking?

4. Data should not be cleaned in Excel - there is no trail of the corrections made and it is too easy to make other mistakes. Instead, data should be cleaned using coding statement in STATA so that there is a clear trail of the modifications made.

5. All data should be available following publication of the results.

7. PLOS authors have the option to publish the peer review history of their article (what does this mean?). If published, this will include your full peer review and any attached files.

Reviewer #1: No

Reviewer #2: No

---

## [Author Response · Author response to Decision Letter 1]

26 Oct 2021

Dear Reviewers

We thank you for the comments and suggestions 

We have seen the your comments carefully one by one. The comments are very essential. We prepared a response document for your comments and revised the protocol manuscript. Thank you.

---

## [Editor Report · Decision Letter 2]

10 Dec 2021

Protocol of a Parallel Group Randomized Control Trial (RCT) for Mobile-assisted Medication Adherence Support (Ma-MAS) intervention among Tuberculosis Patients

PONE-D-21-07609R2

Dear Dr. Sahile,

We’re pleased to inform you that your manuscript has been judged scientifically suitable for publication and will be formally accepted for publication once it meets all outstanding technical requirements.

Kind regards,

Kathleen Finlayson

Academic Editor

PLOS ONE
---

## [Editor Report · Acceptance letter]

16 Dec 2021

PONE-D-21-07609R2 

Protocol of a Parallel Group Randomized Control Trial (RCT) for Mobile-assisted Medication Adherence Support (Ma-MAS) intervention among Tuberculosis Patients 

Dear Dr. Sahile:

I'm pleased to inform you that your manuscript has been deemed suitable for publication in PLOS ONE. Congratulations! Your manuscript is now with our production department. 

Kind regards, 

on behalf of

Dr. Kathleen Finlayson 

Academic Editor

PLOS ONE